# Adverse outcomes after partner bereavement in people with reduced kidney function: Parallel cohort studies in England and Denmark

**Patrick Bidulka**[1]☉*, **Søren Viborg Vestergaard**[2]☉, **Admire Hlupeni**[1], **Anders Kjærsgaard**[2], **Angel Y. S. Wong**[1], **Sinéad M. Langan**[1], **Sigrun Alba Johannesdottir Schmidt**[2,3], **Susan Lyon**[4], **Christian Fynbo Christiansen**[2], **Dorothea Nitsch**[1]

1 Department of Non-Communicable Disease Epidemiology, London School of Hygiene & Tropical Medicine, London, United Kingdom, 2 Department of Clinical Epidemiology, Aarhus University Hospital, Aarhus, Denmark, 3 Department of Dermatology, Aarhus University Hospital, Aarhus, Denmark, 4 Kidney Transplant Recipient, and Widow of Kidney Transplant Recipient, London, United Kingdom

☉ These authors contributed equally to this work.
* patrick.bidulka1@lshtm.ac.uk

**Data Availability Statement:** According to Danish legislation, our approvals to use the Danish data sources for the current study do not allow us to

## Abstract

### Objectives

To investigate whether partner bereavement is associated with adverse cardiovascular and kidney-related events in people with reduced kidney function.

### Design

Two parallel matched cohort studies using linked routinely collected health data.

### Setting

England (general practices and hospitals using linked Clinical Practice Research Datalink, Hospital Episode Statistics, and Office of National Statistics) and Denmark (hospitals and community pharmacies using the Danish National Patient, Prescription and Education Registries and the Civil Registration System).

### Participants

Bereaved people with reduced kidney function (estimated glomerular filtration rate (eGFR) <60mL/min/1.73m$^2$ (England) or hospital-coded chronic kidney disease (Denmark)) and non-bereaved people with reduced kidney function similarly defined, matched on age, sex, general practice (England), and county of residence (Denmark) and followed-up from the bereavement date of the exposed person.

### Main outcome measures

Cardiovascular disease (CVD) or acute kidney injury (AKI) hospitalization, or death.

distribute or make patient data directly available to other parties. Data access may be applied for at the Statistics Denmark. The authors do not have special access privileges to these data. Approvals to use the English data sources for the current study also do not allow us to distribute or make patient data directly available to other parties. Data access may be applied for through the Clinical Practice Research Datalink (CPRD), https://www.cprd.com/. Codelists used to identify outcomes in both England and Denmark, and covariates in England are published on LSHTM data compass (https://doi.org/10.17037/DATA.00002263). Codelists to identify covariates in Denmark are listed in S6–S7 Methods. We cannot disseminate study results directly to study participants, since we used de-identified data in both England and Denmark.

**Funding:** The Danish analyses were partly funded by the Beckett Foundation. SML is funded by a Wellcome Senior Clinical Fellowship in Science (205039/Z/16/Z). For the purpose of Open Access, the author has applied a CC BY public copyright license to any Author Accepted Manuscript (AAM) version arising from this submission. The funders had no role in study design, data collection and analysis, decision to publish, or preparation of the manuscript.

**Competing interests:** I have read the journal's policy and the authors of this manuscript have the following competing interests: All authors have completed an ICJME disclosure form. AH, PB, AK, AYSW, SAJS declare no competing interests, including relevant financial interests, activities, relationships or affiliations. SVV was supported by The Beckett Foundation by a grant administered by Aarhus University which supported this manuscript. DN is the UK Renal Association Director of Informatics Research, a member of the steering group for two GlaxoSmithKline funded studies of kidney function in Sub-Saharan Africa and receives funding from the Health Foundation and the Medical Research Council unrelated to the work in this paper. SVV, CFC, and AK are members of Aarhus University Department of Clinical Epidemiology and are involved in studies with funding from various companies as research grants to (and administered by) Aarhus University. SML is funded by a Wellcome Senior Clinical Fellowship in Science (205039/Z/16/Z) and was funded by the European Academy of Dermatology and Venereology (PPRC-2016-019) for previous bereavement-related research. SL has received consulting fees from STAART-AKI Study Group (reviewing patient information sheets) and the NIHR (lay reviewer), has received payments for

## Results

In people with reduced kidney function, we identified 19,820 (England) and 5,408 (Denmark) bereaved individuals and matched them with 134,828 (England) and 35,741 (Denmark) non-bereaved individuals. Among the bereaved, the rates of hospitalizations (per 1000 person-years) with CVD were 31.7 (95%-CI: 30.5–32.9) in England and 78.8 (95%-CI: 74.9–82.9) in Denmark; the rates of hospitalizations with AKI were 13.2 (95%-CI: 12.5–14.0) in England and 11.2 (95%-CI: 9.9–12.7) in Denmark; and the rates of death were 70.2 (95%-CI: 68.5–72.0) in England and 126.4 (95%-CI: 121.8–131.1) in Denmark. After adjusting for confounders, we found increased rates of CVD (England, HR 1.06 [95%-CI: 1.01–1.12]; Denmark, HR 1.10 [95%-CI: 1.04–1.17]), of AKI (England, HR 1.20 [95%-CI: 1.10–1.31]; Denmark HR 1.36 [95%-CI: 1.17–1.58]), and of death (England, HR 1.10 [95%-CI: 1.05–1.14]; Denmark HR 1.20 [95%-CI: 1.15–1.25]) in bereaved compared with non-bereaved people.

## Conclusions

Partner bereavement is associated with an increased rate of CVD and AKI hospitalization, and death in people with reduced kidney function. Additional supportive care for this at-risk population may help prevent serious adverse events.

## Background

Reduced kidney function is common, affecting at least 5–8% of people of all ages in England and Denmark [1–3]. The prevalence of chronic kidney disease (CKD), the formal diagnosis of reduced kidney function, is at least 30% in people over age 75 [4]. CKD is a progressive and complex disease that is associated with increased risk of acute kidney injury (AKI) [5], stroke [6,7], myocardial infarction [8], and heart failure [9,10], It is unknown to what extent an acute stressor, such as partner bereavement, impacts adverse outcomes in this vulnerable population.

Partner bereavement is one of the most stressful acute life events according to the Social Readjustment Scale [11]. Previous observational studies in the general population have shown that it is associated with short-term increased risk of cardiovascular disease (CVD) and death [12–22]. Possible mechanisms for these associations could be explained by stress manifesting through physiological or behavioural changes in people who are bereaved. For example, previous studies observed immunological changes following partner bereavement, particularly in older adults [23,24]. In addition, decreased adherence to treatment recommendations due to the loss of a caregiver or disruption to routine, as well as unhealthy lifestyle changes (e.g. increased intake of unhealthy foods or alcohol) following partner bereavement could explain these associations.

The impact of partner bereavement on kidney-related outcomes and in people with reduced kidney function is not well described. One study observed considerable declines in the kidney function of caregivers in the three months after their partner's move into a nursing home [25]. In addition, people living with kidney disease have been described as needing more bereavement counselling than those living with other diseases [26], and that current bereavement support for people with end-stage renal disease (ESRD) was generally perceived as poor [27]. Better evidence quantifying the impact of partner bereavement on adverse outcomes in

medical writing or editing from Kidney Care UK, ERA-EDTA for work unrelated to this manuscript. SL has also received support for travel at ERA-EDTA Congress 2018 and 2019 unrelated to this work. SL is also chair of the UK Renal Association Patients' Council and the Guy's & St Thomas' Kidney Patients' Association.

people living with reduced kidney function would inform the design of improved supportive care for this vulnerable population. This topic is particularly relevant in the context of the coronavirus disease (COVID)-19 pandemic, which has likely increased the number of people experiencing partner bereavement.

We aimed to determine whether bereavement in people with reduced kidney function is associated with an increased risk of CVD, AKI, or death. We used routinely collected health data from the UK (1998–2018) and Denmark (1997–2016) to estimate the rate of CVD, AKI, and death in people with reduced kidney function comparing bereaved people with non-bereaved people.

## Methods

### Study design and setting

We conducted two parallel matched cohort studies using routinely collected health data from England and Denmark.

### Data sources

**England.** We used the Clinical Practice Research Datalink (CPRD) Gold primary care data linked to Hospital Episode Statistics (HES) secondary care data, the Index of Multiple Deprivation (IMD), and the Office for National Statistics (ONS) mortality data. We restricted the United Kingdom (UK) primary care cohort to England only since HES is only available in England. CPRD Gold data are shown to be largely representative of the UK population in terms of age, sex, and ethnicity, and include approximately 7% of the UK population [28]. Further details on these datasets are provided in S1 Methods.

**Denmark.** We used national registries linked at the individual level using a unique personal identifier assigned to all Danish residents. We obtained age, sex, civil, and vital status on every Dane from the Danish Civil Registration System [29]. We collected detailed data on inpatient, outpatient, and emergency visits from the Danish National Patient Registry; [30] prescriptions filled at outpatient pharmacies from the Danish National Prescription Registry; [31] and educational attainment from the Danish Education Registers [32]. Further details on these registries are provided in S2 Methods.

### Study population

**England.** We identified partners using an algorithm previously developed using CPRD data [33] (further details are provided in S3 Methods). We identified people who experienced the death of their partner between 1 January 1998 to 31 July 2018 in our bereaved group. We restricted to those registered for ≥1 year at a General Practice (GP) contributing research quality data to the CPRD. Furthermore, we restricted to bereaved individuals with a serum creatinine (SCr) laboratory test corresponding to an eGFR <60mL/min/1.73m$^2$ recorded by the GP within five years prior to the partner death date. We calculated estimated glomerular filtration rate (eGFR) using the Chronic Kidney Disease Epidemiology Collaboration equation [34] (without ethnicity since ethnicity data are incompletely recorded in CPRD-HES). People with no SCr measurement were excluded, as we presumed they had normal kidney function. We defined the partner death date as the index date used to match a comparison cohort.

Among the couples identified by the partner algorithm, we sampled an unexposed (comparison) cohort of people with a living partner matched on age (within +/- 1 year), sex, and GP with replacement. We excluded those who did not have an eGFR measure <60mL/min/1.73m$^2$ within 5 years prior to the index date of the exposed (bereaved) person to whom they

were matched. We kept a maximum of 10 matched unexposed persons for each exposed person.

**Denmark.** In Denmark, the study was nested in an established population of people who lost a partner during 1997–2016 (bereaved) and their non-bereaved comparisons from the general population, matched 1:10 by age, sex and county of residence [35]. In this population, we identified every bereaved person with hospital-recorded CKD (inpatient or outpatient) before the bereavement date, and matched them 1:10 with replacement to non-bereaved people of the same age (+/- 5 years), sex and county of residence with hospital-recorded CKD before the index date. The bereaved partners were identified using an algorithm developed by Statistics Denmark [33] (further details are provided in S4 Methods).

In both England and Denmark, unexposed individuals were censored and moved to the exposed group if they experienced partner bereavement during follow-up.

## Outcomes

Our primary outcomes were first hospitalizations during follow-up for CVD (composite of heart failure, myocardial infarction, and stroke) or AKI, and death. Secondary outcomes included first hospitalizations for heart failure, myocardial infarction, and stroke individually.

We identified first CVD and AKI hospitalizations using ICD-10 codes in the first or second diagnostic position of the inpatient admission's first episode (England) or as a primary or secondary diagnosis in inpatients or outpatients (Denmark). The admission date was used to define the date of the outcome event. We identified deaths using the death date in ONS, or the death date in CPRD if death date in ONS was missing (England) and the Civil Registration System (Denmark).

We followed each participant from the index date until the earliest of the following: date of outcome, death, date of last data collection from the practice (England), transfer out of the general practice for either member of the couple (England), emigration of either member of the couple (Denmark) or the end of study period (31 July 2018 in England, 31 December 2016 in Denmark). We analysed each outcome independently.

## Covariates

We identified potential confounders using hospitalization data, GP data (England only), and civil registration data (Denmark only). Potential confounders included relevant comorbidities and demographic characteristics (age, sex, and socioeconomic status (SES)). In England, we also identified body-mass index (BMI), alcohol intake, and smoking status as potential confounders (defined as described previously [36] and in S5 Methods). These lifestyle data were not available in the Danish data. In both countries, we obtained information from hospital and GP (England only) data anytime before the index date on previously diagnosed AKI, cerebrovascular disease, chronic obstructive pulmonary disease, diabetes, hypertension, myocardial infarction, other ischaemic heart disease, peripheral artery disease, connective tissue diseases, dementia, peptic ulcer, non-haematological malignancies, haematological malignancies, liver disease, and prevalent heart failure. In Denmark, diabetes was defined as either a hospital diagnosis or a filled prescription for an antidiabetic drug. In England, we used the most recent eGFR recorded in primary care to categorise baseline CKD stage according to cut-points from the Kidney Disease Improving Global Outcomes guidelines (data unavailable in Denmark) [37]. These categories were CKD stage 3a (eGFR 45-59mL/min/1.73m$^2$), CKD stage 3b (eGFR 30-44mL/min/1.73m$^2$), and CKD stages 4–5 (eGFR 0-30mL/min/1.73m$^2$). In Denmark, duration of CKD was defined as time since first CKD diagnosis at index date. eGFR data were not

available in the data sources we used in Denmark. As a proxy for SES, we used IMD quintiles (England) or highest educational attainment (Denmark).

### Statistical analysis

We summarised baseline characteristics and absolute rates per 1,000 person-years (PY) for each outcome by exposure status (bereaved or non-bereaved) in both countries. We then used Cox proportional hazards models to calculate unadjusted hazard ratios (HR) with 95% confidence intervals (CI) for each outcome stratified by matched sets to account for the matching factors. In an adjusted model, we then added comorbidities, history of AKI, SES, and lifestyle factors (England only) as covariables. We used a complete case approach since the missing data (on lifestyle variables in England and educational attainment in Denmark) are unlikely to be missing at random with respect to the outcome and therefore multiple imputation would be invalid [38]. We stratified results for primary outcomes by age group (<64 years, 65–74 years, and 75+ years), sex, prevalent CVD (for the CVD outcome only), and CKD stage (England only) and presented the stratified HR and 95% CI for each category. We specified all analyses a priori. We assessed proportionality by visual inspection of log-log plots.

We conducted two sensitivity analyses in the English cohort to assess the robustness of our results. First, we shortened the study period to 1 January 2010–31 July 2018 since AKI coding was poor prior to 2010 [39]. Second, we repeated the main analysis but matched bereaved individuals to non-bereaved individuals using matching without replacement. We performed this analysis to explore the impact of not accounting for the repeated use of unexposed individuals across (but not within) matched sets in the main analysis.

Data management and analyses were performed using Stata version 16 (StataCorp, Texas) in England, and SAS version 9.4 (Cary, NC, USA) in Denmark.

### Patient involvement statement

This study was designed and conducted without patient involvement. A bereaved patient representative (SL) critically reviewed and interpreted the results, and contributed to the writing and editing of the manuscript.

### Ethics

In England, the study was approved by the London School of Hygiene and Tropical Medicine Research Ethics Committee (Reference: 16545) and by the CPRD Independent Scientific Advisory Committee (ISAC Protocol Number: 19_034). We did not obtain informed consent since these data are de-identified. GPs opt-in to sharing de-identified patient data and individual patients can opt-out. In Denmark, the study was reported to the Danish Data Protection Agency through registration at Aarhus University (record number 2016-051-000001/812). Danish legislation does not require approval by an ethical review board or informed consent from patients for registry-based studies.

## Results

### Baseline characteristics

In England, we identified 19,820 bereaved people with reduced kidney function and matched them with 134,828 non-bereaved people with reduced kidney function. In the Danish population of bereaved people, we identified 5,408 bereaved people with hospital-diagnosed CKD and matched them to 35,741 non-bereaved comparisons with CKD (**Fig 1**).

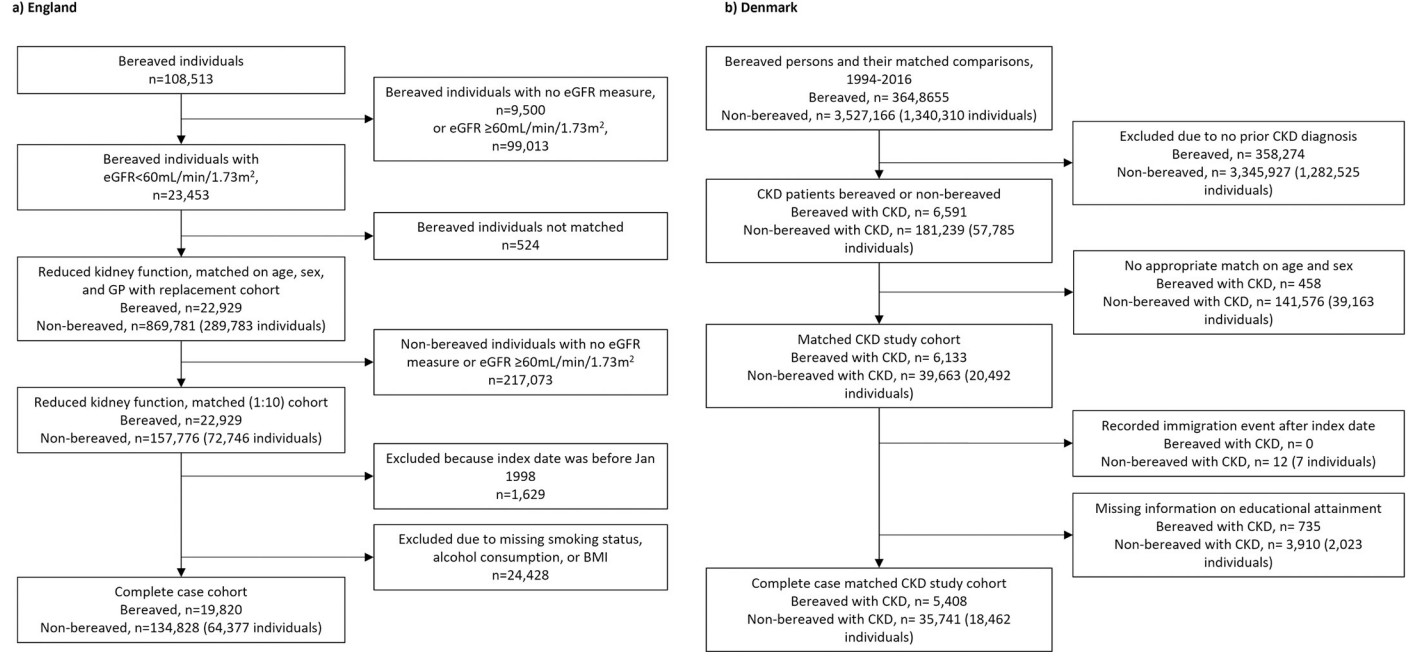

**Fig 1.** Flow diagram of cohort sampling in England (1a) and Denmark (1b).

We observed equal distribution of sex and age between exposure groups in both cohorts since we matched on these variables; however, the median age and proportion of females were higher in England than Denmark (**Table 1**). Most participants in England had eGFR corresponding to CKD stage 3a and CKD stages were equally distributed in bereaved and non-bereaved people. Duration of CKD at index date in the Danish cohort was slightly lower in bereaved than in non-bereaved. Hypertension, ischaemic heart disease, non-haematological malignancy, and diabetes were the most common comorbidities (**Table 1**), and comorbidity prevalence was well balanced between bereaved and non-bereaved groups in both countries.

In England, there was a slightly higher proportion of current smokers in the bereaved (14.6%) versus non-bereaved group (12.3%), and a slightly lower proportion of current drinkers in the bereaved (68.8%) versus non-bereaved group (72.6%). However, the prevalence of smoking- and alcohol-related comorbidities, such as peptic ulcer and cardiovascular disorders, was similar in bereaved and non-bereaved in both countries. In England, most participants were overweight or obese (62.8% of bereaved, 66.1% of non-bereaved people). The bereaved group had a higher proportion of people in the most deprived IMD quintile (13.9% of bereaved vs. 11.4% of non-bereaved) although both groups had over-representation of people in the least deprived quintiles. In Denmark, education level was slightly lower in bereaved than non-bereaved people (**Table 1**).

## Cardiovascular outcome

In bereaved people, we observed CVD hospitalization rates of CVD of 31.7 per 1,000 person-years (95%-CI: 30.5–32.9) in England, and of 78.8 per 1,000 person-years (95%-CI: 74.9–82.9) in Denmark. Compared with non-bereaved people with reduced kidney function, the adjusted HR of CVD in bereaved was 1.06 (95%-CI: 1.01–1.12) in England and 1.10 (95%-CI: 1.04–1.17) in Denmark. In both countries, the rate of heart failure was higher than the rate of myocardial infarction and stroke. In England, only heart failure was associated with partner

**Table 1.** Baseline characteristics of people with CKD who were bereaved and their non-bereaved matched comparisons in England (1998–2018) and Denmark (1997–2016).

| | England | | Denmark | |
|---|---|---|---|---|
| | **Bereaved** | **Comparison** | **Bereaved** | **Comparison** |
| **Overall, n (%)** | 19,820 | 134,828 | N = 5,408 | N = 35,741 |
| **Sex, n (%)** | | | | |
| Male | 6,809 (34.4) | 46,469 (34.5) | 2,498 (46.2) | 17,411 (48.7) |
| Female | 13,011 (65.6) | 88,359 (65.5) | 2,910 (53.8) | 18,330 (51.3) |
| **Age in years at index date[1], median (IQR)** | 80 (75;84) | 81 (76;85) | 75.4 (68.5;81.3) | 76.3 (69.8;81.7) |
| **Age groups in years at index date[1]** | | | | |
| <65 | 525 (2.6) | 2,803 (2.1) | 882 (16.3) | 4,439 (12.4) |
| 65–74 | 3,846 (19.4) | 28,883 (21.4) | 1,723 (31.9) | 11,322 (31.7) |
| 75+ | 15,449 (77.9) | 103,142 (76.5) | 2,803 (51.8) | 19,980 (55.9) |
| **Years since CKD diagnosis, median (IQR)** | NA | NA | 4.5 (1.8;9.9) | 4.4 (1.7;10.0) |
| **CKD stage (based on last recorded eGFR before index date[1])** | | | | |
| Stage 3a | 13,648 (68.9) | 95,989 (71.2) | NA | NA |
| Stage 3b | 4,967 (25.1) | 31,910 (23.7) | NA | NA |
| Stage 4 | 1,048 (5.3) | 6,031 (4.5) | NA | NA |
| Stage 5 | 157 (0.8) | 898 (0.7) | NA | NA |
| **Hospital-diagnosed acute kidney injury prior to index date[1], n (%)** | 740 (3.7) | 3,830 (2.8) | 370 (6.8) | 2,452 (6.9) |
| **Any Renal replacement therapy (RRT), n (%)** | 76 (0.4) | 602 (0.4) | 679 (12.6) | 4,342 (12.1) |
| **Type of RRT, n (%)** | | | | |
| Kidney transplant | NA | NA | 60 (1.1) | 420 (1.2) |
| Acute dialysis | NA | NA | 290 (5.4) | 1,828 (5.1) |
| Chronic dialysis | NA | NA | 329 (6.1) | 2,094 (5.9) |
| None | 19,744 (99.6) | 134,226 (99.6) | 4,729 (87.4) | 31,399 (87.9) |
| **Hospital-diagnosed comorbidity[2], n (%)** | | | | |
| Myocardial infarction | 2,307 (11.6) | 14,945 (11.1) | 889 (16.4) | 5,829 (16.3) |
| Congestive heart failure | 2,852 (14.4) | 17,522 (13.0) | 1,088 (20.1) | 7,163 (20.0) |
| Peripheral vascular disease | 1,618 (8.2) | 10,139 (7.5) | 1,033 (19.1) | 6,811 (19.1) |
| Cerebrovascular disease | 3,114 (15.7) | 21,053 (15.6) | 1,132 (20.9) | 7,961 (22.3) |
| Dementia | 714 (3.6) | 4,665 (3.5) | 169 (3.1) | 1,245 (3.5) |
| Chronic pulmonary disease | 2,108 (10.6) | 12,801 (9.5) | 915 (16.9) | 5,922 (16.6) |
| Connective tissue disease | 1,852 (9.3) | 11,900 (8.8) | 476 (8.8) | 3,450 (9.7) |
| Peptic ulcer disease | 1,838 (9.3) | 12,050 (8.9) | 641 (11.9) | 3,975 (11.1) |
| Liver disease | 198 (1.0) | 1,296 (1.0) | 161 (3.0) | 953 (2.7) |
| Diabetes | 4,432 (22.4) | 27,896 (20.7) | 1,976 (36.5) | 12,473 (34.9) |
| Non-haematological malignancy | 4,500 (22.7) | 30,225 (22.4) | 1,123 (20.8) | 7,882 (22.1) |
| Haematological malignancy | 280 (1.4) | 1,931 (1.4) | 110 (2.0) | 892 (2.5) |
| Hypertension | 14,565 (73.5) | 98,876 (73.3) | 3,134 (58.0) | 21,136 (59.1) |
| Ischaemic heart disease | 6,032 (30.4) | 40,125 (29.8) | 1,851 (34.2) | 12,461 (34.9) |
| **Smoking status, n (%)** | | | | |
| Non-smoker | 6,012 (30.3) | 41,755 (31.0) | NA | NA |
| Ex-smoker | 10,914 (55.1) | 76,543 (56.8) | | |
| Current smoker | 2,894 (14.6) | 16,530 (12.3) | NA | NA |
| **Alcohol intake, n (%)** | | | | |
| Non-drinker | 2,509 (12.7) | 14,296 (10.6) | NA | NA |
| Ex-drinker | 3,665 (18.5) | 22,642 (16.8) | NA | NA |
| Current drinker | 13,646 (68.8) | 97,890 (72.6) | NA | NA |

*(Continued)*

**Table 1.** (Continued)

| | England | | Denmark | |
|---|---|---|---|---|
| | **Bereaved** | **Comparison** | **Bereaved** | **Comparison** |
| **Body mass index (kg m⁻²), n (%)** | | | | |
| Underweight (<18.5) | 472 (2.4) | 2,291 (1.7) | NA | NA |
| Normal weight (18.5–24.9) | 6,897 (34.8) | 43,381 (32.2) | NA | NA |
| Overweight (25–29.9) | 7,625 (38.5) | 54,907 (40.7) | NA | NA |
| Obese (≥30) | 4,826 (24.3) | 34,249 (25.4) | NA | NA |
| **Index of multiple deprivation, n (%)** | | | | |
| 1 (least deprived) | 4,512 (22.8) | 33,964 (25.2) | NA | NA |
| 2 | 4,629 (23.4) | 34,100 (25.3) | NA | NA |
| 3 | 4,375 (22.1) | 29,261 (21.7) | NA | NA |
| 4 | 3,548 (17.9) | 22,117 (16.4) | NA | NA |
| 5 (most deprived) | 2,756 (13.9) | 15,386 (11.4) | NA | NA |
| **Educational attainment (years), n (%)** | | | | |
| Short (7–10) | NA | NA | 2,994 (55.4) | 17,571 (49.2) |
| Medium (11–12) | NA | NA | 1,816 (33.6) | 13,069 (36.6) |
| Long (≥13) | NA | NA | 598 (11.1) | 5,101 (14.3) |

[1]Index date is the bereavement date for the bereaved individual. This same date is the index date for all non-bereaved people within the matched set.

[2]Comorbidities identified using ICD-10 codes in hospital data (England and Denmark) recorded any time prior to the index date. Read codes recorded by the GP anytime prior to the index date were also used in England.

CKD: Chronic kidney disease, eGFR: Estimated glomerular filtration, IQR: Interquartile range.

bereavement (HR of 1.08 [95%-CI: 1.00–1.17]), whereas heart failure, myocardial infarction, and stroke were associated with bereavement in Denmark (**Table 2**). The increased HR of CVD associated with bereavement was observed in both sexes in Denmark, while it was observed in men only in England (**Fig 2** and **S1 Table**). Furthermore, the increased CVD relative risk in bereaved people was greatest in younger age groups (**Fig 2** and **S1 Table**). When stratifying by CKD stage in England, there was evidence of greater risk of CVD in bereaved vs. non-bereaved people with stage 3a (HR 1.10 [95%-CI: 1.03–1.17]), while there was no evidence of an increased risk in those with stage 3b or stages 4–5 (**Fig 2** and **S1 Table**).

## AKI outcome

Rates of hospital-recorded AKI in the bereaved groups with reduced kidney function were comparable in England and Denmark (13.2 per 1,000 person-years [95%-CI: 12.5–14.0] in England, 11.2 per 1,000 person-years [95%-CI: 9.9–12.7] in Denmark). Compared with non-bereaved people, bereaved people had a higher risk of AKI with adjusted HRs of 1.20 (95%-CI: 1.10–1.31) in England and 1.36 (95%-CI: 1.17–1.58) in Denmark (**Table 2**). There were no clear differences in HRs of AKI between subgroups of age and sex in either setting. In England, there was no evidence of an increased risk of AKI in bereaved vs. non-bereaved people for those with CKD stages 4–5. Subgroups with eGFR 45–59 and 30-44mL/min/1.73m² had similar increased risks of AKI in the bereaved compared with the non-bereaved groups (HR 1.22 [95%-CI: 1.10–1.36] and HR 1.20 [95%-CI: 1.04–1.38], respectively) (**Fig 2** and **S1 Table**).

## Mortality outcome

The mortality rate in bereaved persons with reduced kidney function in England (70.2 per 1,000 person-years [95%-CI: 68.5–72.0]), was lower than that in bereaved CKD patients in

**Table 2. Risk of CVD, AKI, and death in person with CKD with or without bereavement in two different populations.**

| Population | Outcome | Bereaved cohort | | | Comparison cohort | | | Unadjusted HR (95% CI) | Adjusted HR (95% CI) |
|---|---|---|---|---|---|---|---|---|---|
| | | Number of events | Person years at-risk | Rate per 1,000 person-years | Number of events | Person years at-risk | Rate per 1,000 person-years | | |
| England | **Composite CVD** | 2621 | 82747 | 31.7 (30.5–32.9) | 14942 | 538165 | 27.8 (27.3–28.2) | 1.06 (1.01–1.11) | 1.06 (1.01–1.12) |
| | Heart failure | 1424 | 85102 | 16.7 (15.9–17.6) | 7827 | 551864 | 14.2 (13.9–14.5) | 1.09 (1.02–1.16) | 1.08 (1.00–1.17) |
| | Myocardial infarction | 695 | 85943 | 8.09 (7.51–8.71) | 4012 | 556262 | 7.21 (6.99–7.44) | 1.07 (0.98–1.17) | 1.03 (0.94–1.13) |
| | Stroke | 846 | 85993 | 9.84 (9.20–10.5) | 5027 | 556491 | 9.03 (8.79–9.29) | 0.99 (0.92–1.08) | 1.01 (0.93–1.10) |
| | **AKI** | 1136 | 85950 | 13.2 (12.5–14.0) | 5560 | 557977 | 9.96 (9.71–10.2) | 1.18 (1.10–1.27) | 1.20 (1.10–1.31) |
| | **Death** | 6135 | 87389 | 70.2 (68.5–72.0) | 31194 | 564437 | 55.3 (54.7–55.9) | 1.12 (1.08–1.15) | 1.10 (1.05–1.14) |
| Denmark | **Composite CVD** | 1494 | 18962 | 78.8 (74.9–82.9) | 8265 | 110315 | 74.9 (73.3–76.6) | 1.12 (1.06–1.18) | 1.10 (1.04–1.17) |
| | Heart failure | 898 | 20364 | 44.1 (41.3–47.1) | 5147 | 117152 | 43.9 (42.7–45.1) | 1.10 (1.02–1.18) | 1.09 (1.01–1.18) |
| | Myocardial infarction | 414 | 21333 | 19.4 (17.6–21.3) | 2213 | 122166 | 18.1 (17.4–18.9) | 1.11 (1.00–1.23) | 1.06 (0.94–1.19) |
| | Stroke | 558 | 20989 | 26.6 (24.4–28.9) | 2811 | 120888 | 23.3 (22.4–24.1) | 1.17 (1.07–1.28) | 1.15 (1.04–1.27) |
| | **AKI** | 246 | 21925 | 11.2 (9.9–12.7) | 1157 | 125022 | 9.3 (8.7–9.8) | 1.40 (1.21–1.62) | 1.36 (1.17–1.58) |
| | **Death** | 2809 | 22229 | 126.4 (121.8–131.1) | 13956 | 126523 | 110.3 (108.5–112.1) | 1.20 (1.15–1.25) | 1.20 (1.15–1.25) |

*England: adjusted for comorbidities (CKD stage, cerebrovascular disease, heart failure, chronic obstructive pulmonary disease, diabetes, hypertension, ischaemic heart disease, myocardial infarction, peripheral artery disease, connective tissue disease, dementia, peptic ulcers, non-haematological cancer, haematological cancer, liver disease), history of AKI, smoking status, alcohol consumption, BMI category, IMD category.

*Denmark: adjusted for comorbidities (cerebrovascular disease, heart failure, chronic obstructive pulmonary disease, diabetes, hypertension, ischaemic heart disease, myocardial infarction, peripheral artery disease, connective tissue disease, dementia, peptic ulcers, non-haematological cancer, haematological cancer, liver disease), history of AKI, and educational attainment.

AKI: Acute kidney injury, CVD: Cardiovascular disease, CI: Confidence interval, HR: Hazard ratio.

Denmark (126.4 per 1,000 person-years [95%-CI: 121.8–131.1]). However, the risk of death was increased in the bereaved compared with the non-bereaved in both countries (HR 1.10 [95%-CI: 1.05–1.14] in England; HR 1.20 [95%-CI: 1.15–1.25] in Denmark). We found no substantial differences in HRs of death stratified by subgroups of age or sex. In England, there was no evidence of an increased risk of death in bereaved compared with non-bereaved people with CKD stages 4–5 (HR 1.00 [95%-CI: 0.88–1.13]). Subgroups with CKD stages 3a and 3b had similar increased risks of death in the bereaved compared with the non-bereaved groups (HR 1.12 [95%-CI: 1.07–1.16] and HR 1.09 [95%-CI: 1.02–1.16], respectively) (**Fig 2** and **S1 Table**).

## Sensitivity analyses

We found no substantial changes in our results when restricting to years 2010–2018 or when sampling matched comparators without replacement in England (**S2 and S3 Tables**). Hazard ratios were greatest in the first year of follow-up, and diminished with increasing periods of follow-up (**S4–S6 Tables**).

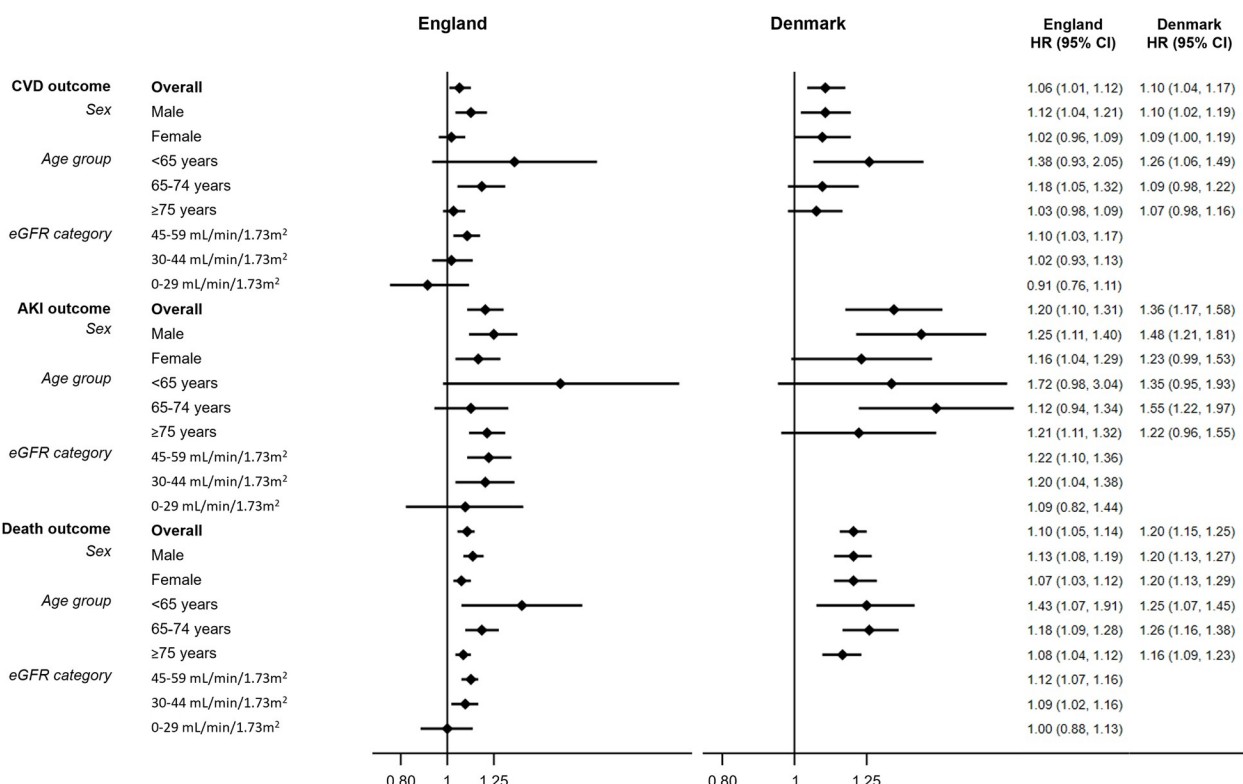

**Fig 2. Forest plot showing adjusted HRs of CVD, AKI and death in bereaved persons with CKD compared to non-bereaved persons with CKD by sex, age-group, CKD stage (England only).**

## Discussion

In people with reduced kidney function, partner bereavement was associated with an increased risk of CVD hospitalization, AKI hospitalization, and death in both England and Denmark. The absolute risk of CVD and death was higher in Danish bereaved patients with hospital-diagnosed CKD compared with English bereaved patients with reduced kidney function in primary care. We observed slightly higher relative risk estimates for all outcomes in Denmark compared to England.

Our aim was to explore the hypothesis that partner bereavement in people with reduced kidney function increased the risk of adverse CVD and kidney-related events, and death. This question is of particular importance in the context of the COVID-19 pandemic; elderly people already at increased risk of living with reduced kidney function [1] are likely to be at higher risk of experiencing partner bereavement due to the pandemic, since COVID-19 mortality increases with age [40]. Furthermore, pandemic-related stressors such as the recommendation to shield by the UK government for people with CKD stage 5, and dialysis or transplant recipients make it more difficult to deal with the practicalities of the death of a partner, and may lead to worse outcomes. Quantifying the increased risk of adverse events, including death, associated with this likely increasingly prevalent exposure may encourage healthcare providers to consider the impact of partner bereavement on high-risk populations during and after the pandemic.

This is the first study to our knowledge to investigate the effect of partner bereavement on adverse outcomes specifically in people with reduced kidney function. We showed consistent results in two countries which strengthens the internal validity of our study. For example,

residual confounding by baseline smoking status, alcohol intake, and BMI are unlikely to account for the observed associations in Denmark since adjusting for these variables in England did not account for the observed associations in this setting. By triangulating data within and between two countries using routinely collected healthcare datasets, we were able to study clinically important outcomes associated with partner bereavement in people with reduced kidney function. We did not meta-analyse these results since the study populations were clinically heterogeneous, as patients identified by hospital-diagnosed CKD (Danish cohort) were younger and had more comorbidities than those with one eGFR measure <60mL/min/1.73m$^2$ measured in primary care (English cohort) [3].

Our study has some limitations. Residual confounding is possible in both settings due to imperfect measurement of covariates as well as unmeasured confounders like social network or diet. Unmeasured confounding by these lifestyle risk factors may partly explain the more pronounced adjusted HRs in Denmark, yet we did adjust for alcohol- and smoking-related diseases and educational attainment to minimise such confounding.

We may have missed couples in England since we relied on a less sensitive algorithm to identify partners compared to Denmark. However, because we used the same methods for identifying bereaved and non-bereaved groups, we do not believe this would have affected our measures of association. In addition, when restricting to people with available information on highest educational attainment in Denmark, we primarily excluded people born before 1945 as the educational registries are virtually complete for people born after 1945 [32].

We did not exclude people with histories of CVD or AKI, meaning it is possible these prevalent conditions were recorded as secondary diagnoses which we incorrectly classified as incident outcome events. Moreover, we did not include outpatient hospital data and cardiovascular and renal disease audit data in England, such as the Myocardial Ischaemia National Audit Project (MINAP) and the UK Renal Registry (UKRR). Thus, we likely missed CVD and AKI outcomes in England. In particular, detection of myocardial infarction hospitalization has been shown to be improved by combining MINAP and HES data [41]. Excluding these data likely underestimated the incidence of study outcomes and diluted effect estimates in England, and might partly explain why incidence of outcomes were higher in Denmark.

Bereaved people without a caregiver at home may be more likely to present to hospital for heart failure, which may partly explain the increased risk in bereaved vs. non-bereaved people. However, this surveillance bias would not explain the increased relative risk of death in bereaved vs. non-bereaved people.

In England, we found no association between bereavement and outcomes of interest in people with CKD stages 4–5 (eGFR 0-30mL/min/1.73m$^2$). In contrast, we found more pronounced relative risks for all study outcomes in Denmark, where patients were identified through hospital records and thus likely had more advanced CKD on average. However, as we were unable to stratify by eGFR levels, we do not know if stage modified the associations of interest in Denmark as well. It is possible that additional supportive care for people with advanced CKD received in nephrology clinics reduces the relative risk of adverse events after partner bereavement and accounts for the null association in this group. Furthermore, people with advanced kidney disease are generally older and multimorbid and may not experience as significant a change in disease status due to acute stressors like bereavement compared to people with less advanced kidney disease. This may explain why we observed a concentration of the increased risk of adverse outcomes post-partner bereavement in people with less advanced kidney dysfunction in England.

Finally, in our main analysis, we sampled our unexposed groups with replacement. This technique may have resulted in too narrow confidence intervals due to the inclusion of some persons in multiple strata, thus leading to artificial statistical homogeneity. However, our

sensitivity analysis in England showed no change in the interpretation of our results when we re-sampled the comparison cohort without replacement.

Previous studies have shown an increased risk of CVD and mortality in people who experienced partner bereavement compared with those with a living partner [12,13], particularly in the short-term [21]. Our study of people with reduced kidney function thus supports these previous findings overall. Unlike previous studies, the association with CVD was driven by an increased risk of heart failure rather than myocardial infarction. This finding could be explained by poor adherence to medications, in particular diuretics, in people with reduced kidney function after the death of their partner, which in turn could cause fluid retention and ultimately heart failure. Further research is needed to understand possible mechanisms to explain the adverse events associated with bereavement in people with reduced kidney function and the possible benefits of interventions for closer monitoring and support.

In conclusion, we found an increased risk of CVD and AKI hospitalizations, and death in people with reduced kidney function who experience partner bereavement compared with people with a living partner. Further observational research to investigate possible mechanisms of this association, for example poor adherence to prescriptions in bereaved individuals, stress-induced pathophysiology, and loneliness, could identify targets to reduce adverse events in this vulnerable population.

## Supporting information

**S1 Table. Association between partner bereavement and study outcomes stratified by age group, sex, and CKD stage.**
(DOCX)

**S2 Table. Risk of CVD, AKI, and death in persons with CKD with or without bereavement restricting study period to 2010–2018 (England only).**
(DOCX)

**S3 Table. Sensitivity analysis–repeat main analysis in English cohort using matching without replacement.**
(DOCX)

**S4 Table. Risk of CVD in person with CKD with or without bereavement in England and Denmark stratified by follow-up periods.**
(DOCX)

**S5 Table. Risk of AKI in person with CKD with or without bereavement in England and Denmark stratified by follow-up periods.**
(DOCX)

**S6 Table. Risk of death in person with CKD with or without bereavement in England and Denmark stratified by follow-up periods.**
(DOCX)

**S1 Methods. Data sources–England.**
(DOCX)

**S2 Methods. Data sources–Denmark.**
(DOCX)

**S3 Methods. Partner identification–England.**
(DOCX)

**S4 Methods. Partner identification–Denmark.**
(DOCX)

**S5 Methods. Lifestyle risk factor algorithms–England.**
(DOCX)

**S6 Methods. Comorbidity codelists (ICD-8 or ICD-10)–Denmark.**
(DOCX)

**S7 Methods. Renal Replacement Therapy (RRT) codelist–Denmark.**
(DOCX)

## Acknowledgments

This study is based in part on data from the Clinical Practice Research Datalink obtained under licence from the UK Medicines and Healthcare products Regulatory Agency. The data are provided by patients and collected by the NHS as part of their care and support. The interpretation and conclusions contained in this study are those of the authors alone.

## Author Contributions

**Conceptualization:** Dorothea Nitsch.

**Data curation:** Patrick Bidulka, Søren Viborg Vestergaard, Anders Kjærsgaard, Angel Y. S. Wong, Sigrun Alba Johannesdottir Schmidt, Dorothea Nitsch.

**Formal analysis:** Patrick Bidulka, Søren Viborg Vestergaard, Admire Hlupeni, Anders Kjærsgaard.

**Funding acquisition:** Sinéad M. Langan.

**Investigation:** Patrick Bidulka, Søren Viborg Vestergaard, Admire Hlupeni, Anders Kjærsgaard, Angel Y. S. Wong, Sinéad M. Langan, Dorothea Nitsch.

**Methodology:** Patrick Bidulka, Søren Viborg Vestergaard, Admire Hlupeni, Anders Kjærsgaard, Sinéad M. Langan, Sigrun Alba Johannesdottir Schmidt, Christian Fynbo Christiansen, Dorothea Nitsch.

**Project administration:** Patrick Bidulka, Søren Viborg Vestergaard, Anders Kjærsgaard, Dorothea Nitsch.

**Resources:** Søren Viborg Vestergaard.

**Software:** Søren Viborg Vestergaard.

**Supervision:** Søren Viborg Vestergaard, Christian Fynbo Christiansen, Dorothea Nitsch.

**Visualization:** Patrick Bidulka.

**Writing – original draft:** Patrick Bidulka, Søren Viborg Vestergaard.

**Writing – review & editing:** Patrick Bidulka, Søren Viborg Vestergaard, Admire Hlupeni, Anders Kjærsgaard, Angel Y. S. Wong, Sinéad M. Langan, Sigrun Alba Johannesdottir Schmidt, Susan Lyon, Christian Fynbo Christiansen, Dorothea Nitsch.

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
