## [Decision Letter · Decision Letter 0]

12 Jul 2021

PONE-D-21-18339

ADVERSE OUTCOMES AFTER PARTNER BEREAVEMENT IN PEOPLE WITH REDUCED KIDNEY FUNCTION: PARALLEL COHORT STUDIES IN ENGLAND AND DENMARK

PLOS ONE

Dear Dr. Bidulka,

Thank you for submitting your manuscript to PLOS ONE. After careful consideration, we feel that it has merit but does not fully meet PLOS ONE’s publication criteria as it currently stands. Therefore, we invite you to submit a revised version of the manuscript that addresses the points raised during the review process.

The manuscript focuses on a topic of potential interest. The study is well performed but few minor issues need to be addressed. To mention some of them, i) unclear why the authors do not have data of eGFR in their cohort; ii) need to assess whether renal replacement therapy (RRT) affects the outcomes, adjusting it in the COX model; iii) need to provide p-values in Table 1; iv) unclear the new term named incidence rate (IR) of CVD introduced in the Result section (line 222); v) need to provide high-resolution figures.

We look forward to receiving your revised manuscript.

Kind regards,

Giuseppe Remuzzi

Academic Editor

PLOS ONE

Journal Requirements:

"This research was funded in whole or in part by the Wellcome Trust (205039/Z/16/Z)."

"The Danish analyses were partly funded by the Beckett Foundation. SML is funded by a Wellcome Senior Clinical Fellowship in Science (205039/Z/16/Z). For the purpose of Open Access, the author has applied a CC BY public copyright license to any Author Accepted Manuscript (AAM) version arising from this submission. The funders had no role in study design, data collection and analysis, decision to publish, or preparation of the manuscript."

4. Please upload a new copy of Figures 1 and 2 as the detail is not clear. Please follow the link for more information: https://blogs.plos.org/plos/2019/06/looking-good-tips-for-creating-your-plos-figures-graphics/" https://blogs.plos.org/plos/2019/06/looking-good-tips-for-creating-your-plos-figures-graphics/

Reviewers' comments:

Reviewer's Responses to Questions

Comments to the Author

1. Is the manuscript technically sound, and do the data support the conclusions?

Reviewer #1: Yes

2. Has the statistical analysis been performed appropriately and rigorously?

Reviewer #1: Yes

3. Have the authors made all data underlying the findings in their manuscript fully available?

Reviewer #1: Yes

4. Is the manuscript presented in an intelligible fashion and written in standard English?

Reviewer #1: Yes

5. Review Comments to the Author

Reviewer #1: This is an interesting manuscript that investigates the association between partner bereavement and hospitalization of CVD and AKI, and death in people with reduced kidney function. The results suggest that partner bereavement was associated with an increased risk of CVD hospitalization, AKI hospitalization, and death in both England and Denmark. The authors provide a clearly written explanation and rationale of the topic and a description of the participants’ selection and outcome assessment. The methods are well detailed and results are professionally presented, providing new evidence on this topic. The study is well designed and presented, while there were some minor issues that need to be addressed:

1. In Demark cohort, the information of baseline eGFR did not offer, which limits the effect of CKD stages on the outcomes. The selection of participants was based on the hospital-diagnosed CKD. Could you explain why you do not have the data of eGFR in your cohort？

2. Both of the cohorts included the information of RRT. It might be interesting to see whether RRT affects the outcomes. You can adjust it in your Cox model.

3. In Table 1, the p-value is missing, so it is difficult to compare the bereaved and the comparison group. Could you explain why did not show the p-value?

4. In the abstract, when you present the information of the rate of CVD, AKI, and death, please add the unit (1000 person-year?).

5. In the Result section line 222, you introduced a new term named incidence rate (IR) of CVD. Is it the same meaning as the rates of hospitalizations with CVD? It might be confusing, please keep using the same term throughout the manuscript.

6. The Figures are not clearly seen, please add high-resolution figures.

6. PLOS authors have the option to publish the peer review history of their article (what does this mean?). If published, this will include your full peer review and any attached files.

Do you want your identity to be public for this peer review? For information about this choice, including consent withdrawal, please see our Privacy Policy.

Reviewer #1: No

---

## [Author Response · Author response to Decision Letter 0]

6 Aug 2021

We have attached a Word file which contains detailed point-by-point responses to each comment from the editor and reviewers. The file is named "Response to Reviewers". 

We hope the editor and reviewer find these responses satisfactory. Again, thank you for taking the time to critically review our manuscript. We appreciate it.

---

## [Decision Letter · Decision Letter 1]

27 Aug 2021

ADVERSE OUTCOMES AFTER PARTNER BEREAVEMENT IN PEOPLE WITH REDUCED KIDNEY FUNCTION: PARALLEL COHORT STUDIES IN ENGLAND AND DENMARK

PONE-D-21-18339R1

Dear Dr. Bidulka,

We’re pleased to inform you that your manuscript has been judged scientifically suitable for publication and will be formally accepted for publication once it meets all outstanding technical requirements.

**The revised version of the manuscript is improved. The authors have adequately addressed the reviewer’s comments.**

Kind regards,

Giuseppe Remuzzi

Academic Editor

PLOS ONE

Additional Editor Comments (optional):

Reviewers' comments:

Reviewer's Responses to Questions

**Comments to the Author**

1. If the authors have adequately addressed your comments raised in a previous round of review and you feel that this manuscript is now acceptable for publication, you may indicate that here to bypass the “Comments to the Author” section, enter your conflict of interest statement in the “Confidential to Editor” section, and submit your "Accept" recommendation.

Reviewer #1: All comments have been addressed

2. Is the manuscript technically sound, and do the data support the conclusions?

Reviewer #1: Yes

3. Has the statistical analysis been performed appropriately and rigorously? 

Reviewer #1: Yes

4. Have the authors made all data underlying the findings in their manuscript fully available?

Reviewer #1: Yes

5. Is the manuscript presented in an intelligible fashion and written in standard English?

Reviewer #1: Yes

6. Review Comments to the Author

Reviewer #1: The authors adequately addressed all the questions raised by me. The paper has improved. This paper will add value to the existed evidence.

7. PLOS authors have the option to publish the peer review history of their article (what does this mean?). If published, this will include your full peer review and any attached files.

Reviewer #1: No

---

## [Editor Report · Acceptance letter]

14 Sep 2021

PONE-D-21-18339R1 

ADVERSE OUTCOMES AFTER PARTNER BEREAVEMENT IN PEOPLE WITH REDUCED KIDNEY FUNCTION: PARALLEL COHORT STUDIES IN ENGLAND AND DENMARK 

Dear Dr. Bidulka:

I'm pleased to inform you that your manuscript has been deemed suitable for publication in PLOS ONE. Congratulations! Your manuscript is now with our production department. 

Kind regards, 

on behalf of

Prof. Giuseppe Remuzzi 

Academic Editor

PLOS ONE